# RETHINKING 'LANGUAGE-ALIGNMENT' IN HUMAN VISUAL CORTEX WITH SYNTAX MANIPULATION AND WORD MODELS

## ABSTRACT

Recent success predicting human ventral visual system responses to images from large language model (LLM) representations of image captions has sparked renewed interest in the possibility that high-level visual representations are aligned to language. Here, we further explore this possibility using image-caption pairs from the Natural Scenes fMRI Dataset, examining how well language-only representations of image captions predict image-evoked human visual cortical responses, compared to predictions based on vision model responses to the images themselves. As in recent work, we find that unimodal language models predict brain responses in human visual cortex as well as unimodal vision models. However, we find that the predictive power of large language models rests almost entirely on their ability to capture information about the nouns present in image descriptions, with little to no role for syntactic structure or semantic compositionality in predicting neural responses to static natural scenes. We propose that the convergence between language-model and vision-model representations and those of high-level visual cortex arises not from direct interaction between vision and language, but instead from common reference to real-world entities, and the prediction of brain data whose principal variance is defined by common objects in common, non-compositional contexts.

## 1 INTRODUCTION

In order to support object-recognition and other visually-grounded cognitive tasks, the visual system must encode representations that abstract beyond identity-preserving transformations —such as variation in viewpoint, scale, and lighting —while simultaneously providing the discriminative power needed to tell apart thousands of objects, agents, and actions. These high-level visual representations must then interface with higher-level cognitive processes, including language. A fundamental question in cognitive science and visual cognitive neuroscience is how vision and language representations become aligned at this interface, and to what extent visual representations are altered by the handoff from perceptual processing of the sensory input to linguistic abstraction. Such questions have driven foundational research in visual cognitive neuroscience (Huth et al., 2012; Konkle and Oliva, 2012; Weiner and Grill-Spector, 2013; Devereux et al., 2013; Bracci and de Beeck, 2016), but have recently experienced new life with the emergence of multimodal (vision-language) deep neural network models (e.g. CLIP (Radford et al., 2021)) that have produced state-of-the-art results in both canonical computer vision tasks (e.g. image categorization) and in prediction of visual cortical activity evoked by natural images (Wang et al., 2023; Conwell et al., 2023).

The rich history of the debate about what happens at the interface between vision and language ultimately means that 'language-alignment' in the context of cognitive (neuro)science means different things to different people. For some, it evokes classic debates about the nature of how the language we speak shapes what we see (Hussein, 2012; Lupyan et al., 2020); for others, it evokes almost the opposite, exposing how the statistics of our perceptual ecology come to be reflected in the ways we communicate about it (Marjieh et al., 2023). One of the most substantive theoretical claims from recent research is that a primary function of high-level visual cortex may actually be generating full 'semantic scene descriptions' (Doerig et al., 2022) (i.e. a language-like description of the scene). This claim is predicated on two key findings: one, that purely linguistic embeddings from large language models (e.g. GUSE or Google's Universal Sentence Encoder) applied to image captions are capable of predicting the majority of explainable variance in image-evoked high-level visual cortical activity;

two) that natural language descriptions to previously unseen images may be decoded with reasonable accuracy from an embedding model fit directly to image-evoked brain activity. This latter finding adds to a rapidly growing list of works that deploy models (seemingly with great success) to reading out the contents and structure of mental life from neural response patterns (Takagi and Nishimoto, 2022; Luo et al., 2023; Tang et al., 2023a).

One overarching issue with work of this nature, however, is that it remains unclear the extent to which the results are driven by information in the brain itself versus the priors of our models. In using language models to predict visual brain activity, especially, there remains substantial ambiguity as to which aspects of these models capture the structure and content of high-level visual representations, and whether language-alignment *per se* improves the correspondence between language-model representations and high-level visual cortex.

In this work, we address these questions by predicting visual responses to the large-scale human fMRI Natural Scenes Dataset (NSD) (Allen et al., 2022) using a variety of unimodal language, unimodal vision, and multimodal (language-aligned) vision models. Like others, we find that unimodal language models are indeed capable of predicting image-evoked brain activity as accurately as unimodal vision models. We also find, however, that the predictive power of large language models (in this dataset) reduces almost entirely to a basis set of simple nouns in no syntactic order, with little to no role for other parts of speech, or compositional semantics. Applying this intuition to recently proposed 'relative representation' (anchor point embedding) techniques (c.f. Moschella et al., 2022; Maiorca et al., 2024; Norelli et al., 2024), we show that we can even 'hand-engineer' a set of 62 simple words whose relative 'word' coordinates in a multimodal foundation model (CLIP) effectively explain the same amount of variance in the image-evoked brain data as the underlying image features themselves. Taken together, these results suggest language model predictivity of visual cortical activity in the Natural Scenes Dataset may have little to do with language *per se*, and far more to do with the recovery of 'grounded information' from co-occurence statistics. This adds as well to a growing consensus that 'vision' and 'language' – at least as learned by modern artificial intelligence algorithms – are in some sense *already aligned* (Pavlick, 2023; Huh et al., 2024), even in the absence of explicit cross-modal learning.

## 2 RESULTS

Our main experimental assay consists of using features extracted from vision-only, language-only, or hybrid vision-language hierarchical deep learning models, and shallow word-vectorizing models, to predict the representational geometries of voxel responses in the early visual (EVC) and occipitotemporal cortices (OTC) of 4 subjects viewing 1000 MS-COCO images from the 7T fMRI Natural Scenes Dataset (Allen et al., 2022). We split these 1000 images into a training and test set of 500 images each. Language descriptions of these images come in the form of captions (5 per image, provided as part of the COCO metadata).

We employ two metrics of model-to-brain comparison: classical (unweighted) and voxelwise-encoding representational similarity analysis (cRSA and eRSA, respectively) (Kriegeskorte et al., 2008a; Kaniuth and Hebart, 2021; Konkle and Alvarez, 2022). cRSA considers all of the features from a given model equally in computing an image-wise representational similarity matrix (RSM), which is then directly compared with the target EVC or OTC RSM. eRSA involves first fitting voxelwise encoding models from features maps with a combination of sparse random projection (for dimensionality reduction) and ridge regression (with cross-validated lambda hyperparameters for each voxel) The predicted responses from these encoding models are then used to generate a reweighted RSM, which we then compare to the target EVC or OTC RSM[1]. For DNN models, we compute scores across all layers on the training set, and select the most predictive layer for assessment on a held-out test set. All scores we report are the generalization scores of each RSA metric on this held-out test set, with no contamination from selection procedures used on the training set (including layer selection and voxel-encoding hyperparameters).

---

[1]An important methodological note here is that these 'predicted responses' can be directly converted into 'voxel-wise encoding scores': that is, the correlation between predicted and actual responses *per voxel*. In this work, we choose to report only the cRSA and eRSA scores so that they may be directly compared – both in terms of their noise ceilings (GSN) and units (dissimilarity in $1 - r_{Pearson}$).

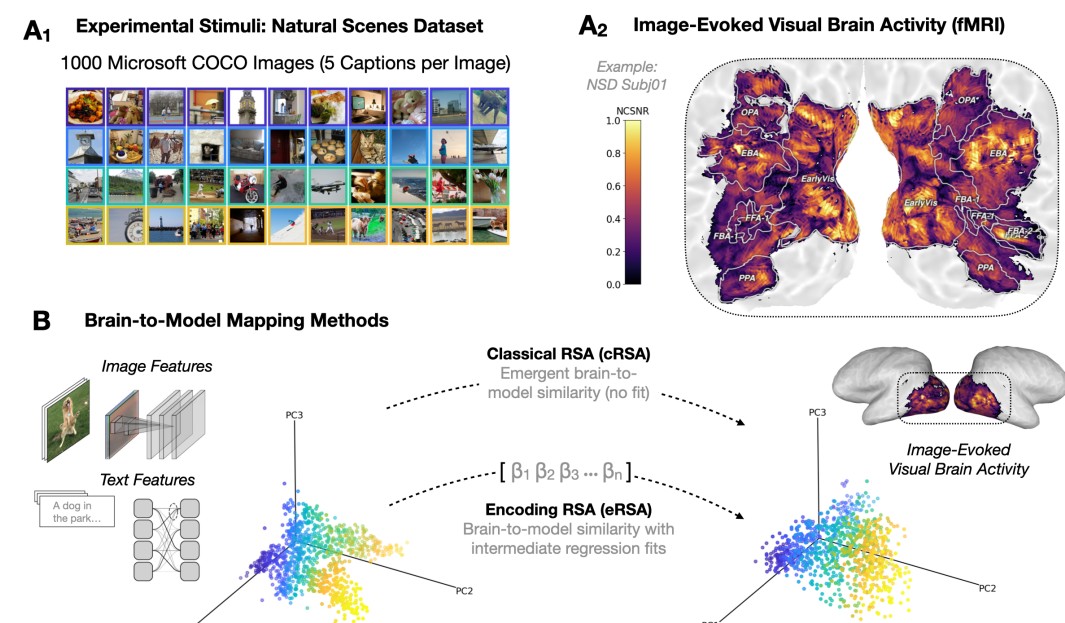

Figure 1: Overview of our experiments. **A** Our primary dataset consists of brain activity in the ventral stream of 4 subjects from the Natural Scenes Dataset viewing 1000 images from the Microsoft COCO (Lin et al., 2014) dataset. Each of these Microsoft COCO images is associated with 5-6 captions provided by human annotators. **B** To predict this activity, we first extract either image or text features from a deep neural network model, using either the images themselves, or the first 5 COCO captions associated with each image. To map these features to brain activity, we use one of two forms of representational similarity analysis (RSA): classical RSA, which enforces fully emergent similarity between a given model's feature space and that of the brain, or, a voxelwise-encoding RSA, which involves first linearly re-weighting model features to predict each of the voxels in a target ROI, and then using held-out test images to generate a model-predicted representational similarity matrix (RSM) for comparison to the true brain RSM.

Finally, we compare these scores to subject- and ROI-specific noise ceilings that are estimated using the Generative Modeling of Signal and Noise (GSN) toolbox Kay et al. (2024). In brief, GSN attempts to model the distinct contributions of signal and noise covariance structure to the observed brain measurements, to then estimate the maximum degree to which a computational model can predict the RSM of a given pool of voxels. These noise ceilings are $r_{Pearson} = 0.69$, $CI_{95} = [0.63, 0.74]$ in EVC and 0.80 [0.73, 0.85] in OTC. An overview of our methodology is available in Figure 1. More details on all our experimental methods, including brain data prepossessing, voxel selection, model selection, feature extraction, brain mapping metrics, and noise ceiling calculations, may be found in the Methods Appendix.

A summary of results for all model comparisons and manipulations is displayed in Table 1. Unless otherwise noted, we use the following convention in the reporting of summary statistics: arithmetic mean [lower 95%, upper 95% bootstrapped confidence interval].

## 2.1 VISION-ONLY VERSUS LANGUAGE-ONLY MODELS

As a primary point of comparison, we consider the relative difference in brain-predictivity of the unimodal models (vision-only versus language-only). The main question we are asking in this analysis is whether these two drastically different model types (the latter of which learns in the absence of visual input) are nonetheless comparable in their ability to predict responses in visual cortex.

The vision-only models in this comparison consist entirely of self-supervised (visual) contrastive learning models, whose learned representations are the product of a training procedure that operates

Table 1: Comparison of Model Types in Predicting OTC Activity

| Analysis Name | Model Type | OTC-Predictive Accuracy [± 95% BCI] | | | |
|---|---|---|---|---|---|
| | | cRSA Score | | eRSA Score | |
| Vision vs Language | Vision-only (mean) | 0.338 | [0.329, 0.348] | 0.682 | [0.662, 0.700] |
| | Language-only (mean) | 0.277 | [0.257, 0.297] | 0.662 | [0.650, 0.675] |
| | Vision-only (max) | 0.384 | [0.362, 0.392] | 0.712 | [0.703, 0.721] |
| | Language-only (max) | 0.437 | [0.414, 0.466] | 0.689 | [0.673, 0.694] |
| Word-Level Models | CountVec (trigrams) | 0.210 | [0.192, 0.221] | 0.584 | [0.556, 0.601] |
| | GLOVE (all words) | 0.320 | [0.300, 0.340] | 0.650 | [0.610, 0.700] |
| | GLOVE (nouns only) | 0.310 | [0.290, 0.330] | 0.630 | [0.590, 0.680] |
| Anchor Point Embeds | CLIP-Vision (768-D) | 0.322 | [0.311, 0.340] | 0.668 | [0.624, 0.734] |
| | 62-word (hypothesis) | 0.320 | [0.305, 0.337] | 0.671 | [0.635, 0.708] |
| | 62-word (random sample) | 0.211 | [0.202, 0.221] | 0.556 | [0.523, 0.590] |

only over individual image instances, and involves no explicit semantic labels (e.g., they do not rely on the one-hot category encoding vectors that define category distinctions in supervised object recognition models).

The language-only models in this comparison consist entirely of transformer-based deep neural network models (Vaswani et al., 2017) trained on one of two tasks: masked language modeling (the prediction of a *masked* token removed at random from an input sequence of tokenized words) (Devlin et al., 2018) or causal language modeling (the prediction of the *next* word following an input sequence of tokenized words) (Radford et al., 2018). The primary operations in these models consist of multi-head attention computations over the tokenized words (plus positional embeddings) of a given sentence. Their learned representations can thus (in both theory and practice) capture transition probabilities and long-range dependencies between different words and concepts.

These model classes learn from different data modalities, under very different task constraints. Remarkably, despite these differences, we find the OTC predictivity of these two model types to be comparable (see Figure 2A): for the eRSA metric, these sets are not significantly different in their average brain predictivity (vision-only mean $r_{Pearson}$ = 0.682 [0.662, 0.700], language-only mean $r_{Pearson}$ = 0.662 [0.65, 0.675]; $p$ = 0.08, $g_{Hedges}$ = -0.92). For the more stringent cRSA metric, the scores of the vision-only models are significantly higher on average than those of the language-only models ($r_{Pearson}$ = 0.338 [0.329, 0.348], mean $r_{Pearson}$ = 0.277 [0.257, 0.297], respectively; ; $p$ = 0.031, $g_{Hedges}$ = -0.915). Worth noting, however, is that there is a high degree of variability in cRSA scores amongst the language-only models, which obscures a striking standout: SBERT-Mini-LM6, a language-only model whose cRSA score is the highest of all the models we survey (mean $r_{Pearson}$ = 0.437 [0.414, 0.466]; the next highest-scoring model in cRSA yields $r_{Pearson}$ = 0.380 [0.369, 0.393]. In other words, the highest-ranking (unweighted) model of the representational geometry of high-level visual cortex in this survey is not a visual model at all, and learns entirely without visual input.

We next compare the prediction of pure-vision and pure-language models in prediction of early visual cortex. Here, we find that the same language-only models that perform comparably with vision-only models in prediction of OTC perform uniformly worse (and by a large margin) in prediction of EVC. The mean accuracy of vision-only models in EVC is $r_{Pearson}$ = 0.295 [0.28, 0.31] in cRSA and 0.48 [0.46, 0.5] in eRSA. The mean accuracy of language-only models is $r_{Pearson}$ = 0.087 [0.081, 0.095] in cRSA and 0.149 [0.14, 0.16] in eRSA. This difference is significant and substantial in both metrics ($p$ = $4.11e^{-21}$, $g_{Hedges}$ = -11.2 in cRSA; $p$ = $7.78e^{-19}$ $g_{Hedges}$ = -32.5 in eRSA). Thus, despite their relative parity in high-level visual cortex, we find that language models yield very little in terms of the representational structure necessary to predict responses in early visual cortex, where voxels are tuned to lower-level image attributes such as oriented lines, edges, and textures.

A summary of these results is available in Figure 2 and Table 1 (Analysis Name: Vision versus Langauge). The results of two follow-up analyses (one that compares the randomly initialized versions of these models to assess for differences in architectural inductive bias, and another that uses

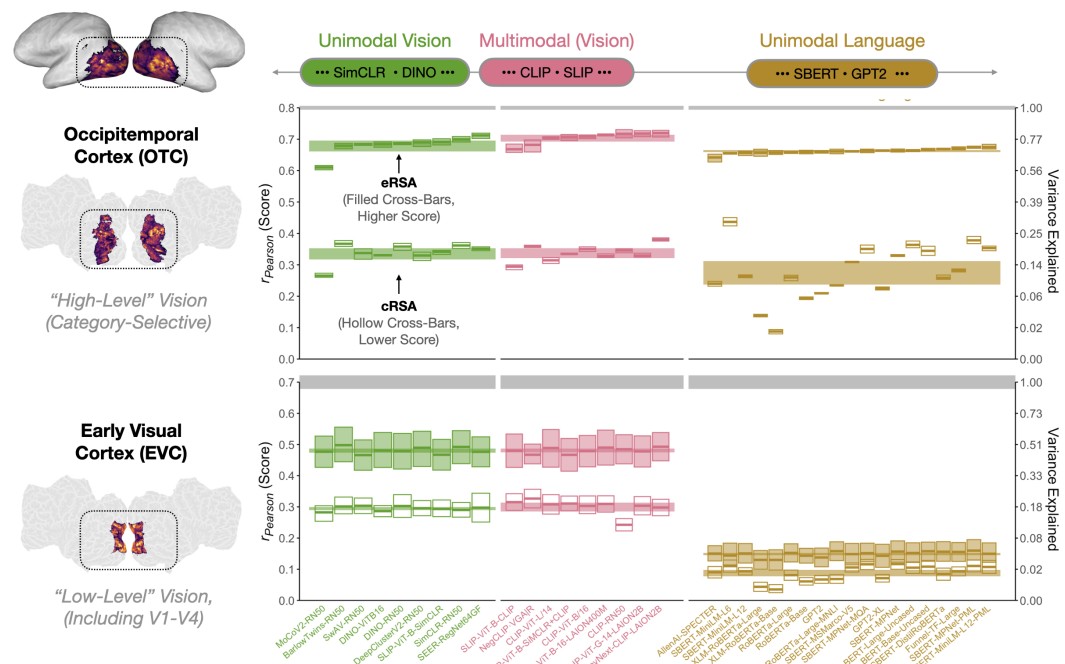

Figure 2: **(Multi)Modal Model Comparison** Performance of a selection of vision-only, language-only, and language-aligned vision models in prediction of early visual and occipitotemporal cortical activity evoked by natural images. Crossbars are each model's mean predictive accuracy $\pm$ grand-mean-centered CI across subjects. (Filled / shaded cross-bars correspond to cRSA scores; hollow / unshaded cross-bars correspond to eRSA scores.). The extended horizontal ribbons are to boot-strapped 95% CIs across model sets. The gray ribbon is the bootstrapped 95% CI of the brain data noise ceiling across subjects. Variance explained is computed as the squared $r_{Pearson}$ score divided by the squared noise ceiling. Models are sorted by their eRSA scores in OTC.

variance partitioning to assess the brain-predictive structure unique to either modality) are available in Appendix A.2.

## 2.2 FROM SENTENCE EMBEDDINGS TO WORD MODELS

So far, the results of our model comparisons have provided two takeaways: first, that pure visual learning and pure language learning may be converging on representations that are equally predic-tive of high-level visual cortex; second, that further modifying purely visual representations with language does not meaningfully increase this predictivity. But what exactly is it about the learned representations of these language models that gives us predictivity of perceptually grounded brain activity?

To answer this question, we first perform a series of experiments in which we manipu-late the natural language inputs to our pure-language models and compare them to far sim-pler word-based NLP models that long predate Transformer-based LLMs in neural analyses (c.f. Huth et al., 2012; Carlson et al., 2014). More specifically, we break down the linguistic de-scriptions of the NSD probe images into increasingly simpler subcomponents (from sentences to phrases, and from phrases to individual words), testing the extent to which these progressive degrada-tions preserve brain predictivity comparable with that of LLMs given complete, unmodified sentences. We test these degradations in 3 models: one of the more representative LLMs (SBERT-MiniLM-6), which learns over sentences; in count vectors, which involve no learning at all; and GLOVE models, which learn only over individual words and without hierarchical attention operations. Results from this experiment, as well as a schematic of our model and syntax manipulations is available in Figure 3 and Table 1 (Analysis Name: Word-Level Models).

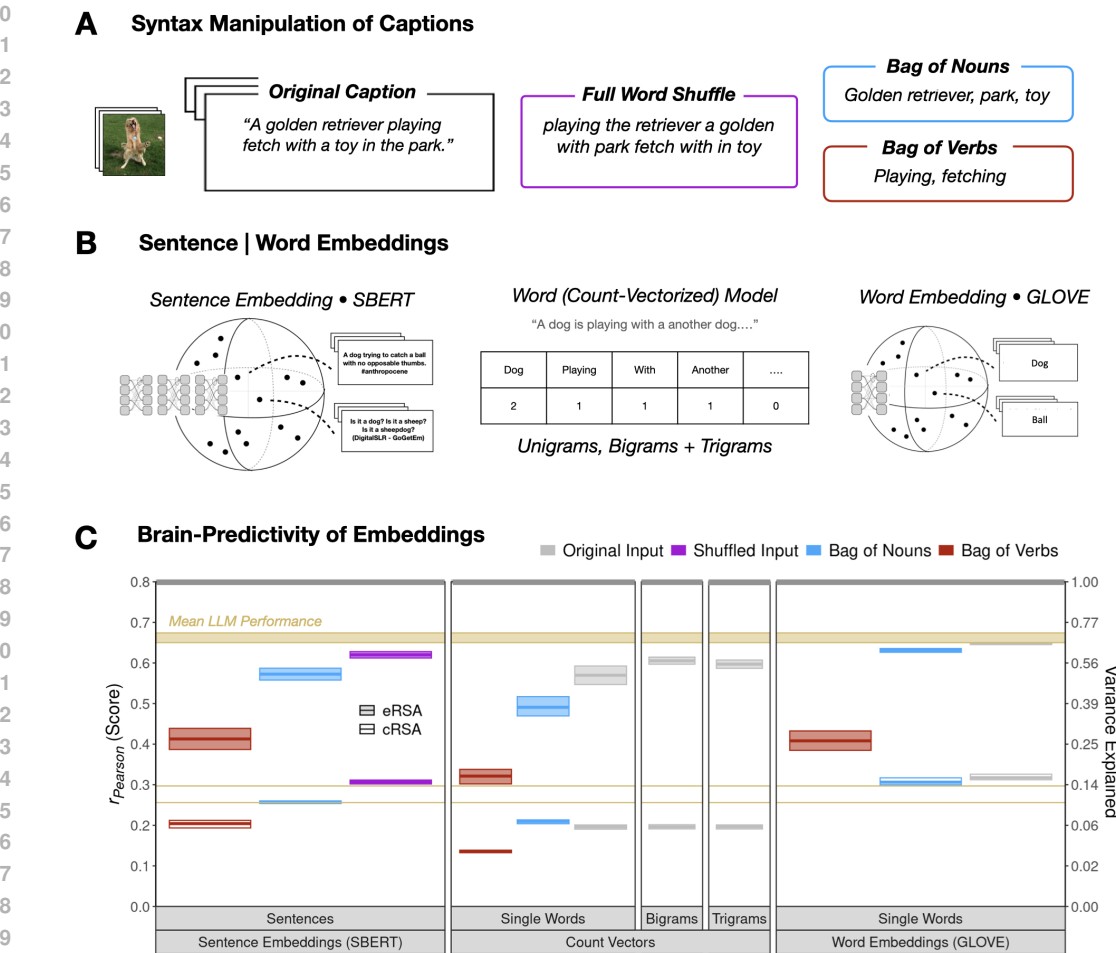

Figure 3: **Syntax Manipulation & Simpler NLP Modeling: A** Examples of the 3 main syntactical manipulations we apply to each caption: full word shuffle (sentence scrambling); bag of nouns; bag of verbs. **B** Schematic of the sentence and word embeddings models we use to 'decompose' the LLM performance: a representative LLM (SBERT-Mini-LM) from our survey of unimodal language models; a word-count vectorized model; and a word embedding model (GLOVE). **C** Results (change in brain predictivity) of the syntax manipulations applied to the input of the 3 different NLP models we assess. Crossbars are the performance of a given model under a certain syntax manipulation (opaque crossbars are the eRSA metric scores; translucent crossbars are the cRSA metric scores .

The lowest performing model in this set of experiments consists of word count vectors computed over the verbs extracted from each image caption (mean $r_{Pearson}$ = 0.135 [0.13, 0.15] in cRSA; 0.32 [0.27, 0.38] in eRSA). The highest performing word-level model consists of the averaged GLOVE word embeddings computed over all individual words in each caption (mean $r_{Pearson}$ = 0.32 [0.30, 0.34] in cRSA; 0.65 [0.61, 0.7] in eRSA. Note already that the performance of this model (which involves no nonlinear hierarchies, and no explicit syntax) is comparable with that of the average LLM (mean $r_{Pearson}$ = 0.277 [0.257, 0.297] in CRSA; 0.662 [0.65, 0.675] in eRSA). The second highest-performing word-level model is the average GLOVE embeddings for nouns only (mean $r_{Pearson}$ = 0.31 [0.29, 0.33] in CRSA; 0.63 [0.59, 0.68] in eRSA). Thus, embeddings from a shallow (log bi-linear) word model (GLOVE), given only nouns, are capable of predicting OTC activity as accurately as LLM embeddings derived from full sentences. Further evidence for the representational significance of nouns may be seen in the performance of the word count model computed over nouns only. This model (unlike GLOVE) does not leverage co-occurence statistics, yet still accounts for

the majority of explainable OTC variance in eRSA (mean $r_{Pearson} = 0.491$ [0.438, 0.564]) and is competitive with mean LLM performance in cRSA ($r_{Pearson} = 0.257$ [0.246, 0.272]).

In sum, this set of experiments suggests that the performance of the language models in predicting visual responses is accounted for almost entirely by the covariance structure instantiated by the nouns of the COCO captions. What exact role the co-occurence statistics learned by a model like GLOVE are playing in these covariance structures remains unclear. What *is* clear is that whatever representations the LLMs are yielding in their prediction of high-level visual cortex, these representations need not be any more sophisticated than a single affine transformation of token-vectorized (embedded) nouns.

## 2.3 'Handcrafted' Word Models by Way of Anchor Point Analysis

Given the relatively high predictive power of the word-level models computed over human-provided captions, we next attempted to construct a simple, hypothesis-driven 'word model' that could effectively capture the variance in occipitotemporal cortex. This process involved two steps. The first step required an element of conjecture to determine which words might sufficiently capture the range of representations evoked by the images in our target stimulus set. (For a real-world example of a similar process, consider the prompt-engineering used in zero-shot evaluations of language-aligned models such as CLIP (Radford et al., 2021)). To score our word model in its prediction of the brain, we used a variant of relative representation analysis (Moschella et al., 2022) (sometimes, and in this work, referred to as anchor point embedding analysis).

Words we included in our hypothesis-driven word model included global descriptors (adjectives applied to whole image, e.g. 'high-resolution' or 'colorful'); agents and objects (e.g. 'man', 'woman', 'child', 'animal', 'vehicle', 'food'); places (e.g. 'desk', 'beach', 'snow', 'desert'); and times-of-day (e.g. 'morning', 'night'). To derive brain-predictivity scores from these prompts, we first generated embeddings for each prompt using CLIP-ResNet50's language encoder, which is a modified RoBERTa architecture. (Note that the use of CLIP is somewhat arbitrary, since this same analysis can be done with *any* algorithm that provides some form of image-text similarity score; see (Maiorca et al., 2024; Norelli et al., 2024)). We then generated the embeddings for each of our target image stimuli, and computed the cosine similarity between each image embedding and all the embeddings associated with our prompts. Finally, we aggregated the resultant image-text similarity matrix together (without softmax) as its own 'feature set', scoring this feature set's brain-predictivity in the same way we scored the feature sets derived from all the models above. As a control, we generated 1000 random samples of unigrams and bigrams from the Brown NLTK corpus (with stimulus counts matching our 'hypothesized' word space), and scored the brain predictivity of these samples using the same encoding pipeline described above.

Strikingly, we found this CLIP-mediated anchor point analysis to perform remarkably well in predicting OTC activity: After some iterative selection (using nested cross-validation scoring on our training set of 500 images), we distilled a 62 word model (consisting only of adjectives and nouns) capable of describing as much variance in the occipitotemporal cortical responses as the underlying CLIP image embeddings (a 768-D vector) from which they were derived: 0.32 [0.305, 0.337] versus 0.322 [0.311, 0.34] in cRSA and 0.671 [0.635, 0.708] versus 0.668 [0.624, 0.734] in eRSA. The mean of the 1000 N=62 random word samples was noticeably lower, but not altogether poor: 0.556 [0.523, 0.59]. A summary of these results is displayed in Figure 4 and Table 1 (Analysis Name: Anchor Point Embeds).

While it might seem odd (or undermining of our hypothesized word space) that these randomly sampled words predicted brain activity so accurately on average, this finding actually strengthens the point this analysis intended to make. Specifically, what sometimes appears to be dense, compositionally complex, or richly structured information in the embedding spaces of large multimodal foundation models can often be reduced to far simpler basis sets. These simpler sets work so long as they provide sufficient coverage of the (representational) variance we are trying to explain. Since the Natural Scenes Dataset is composed of COCO images, its major axes of variance capture the objects and scene attributes found in each of its images. Models that sufficiently capture word co-occurrence appear to predict these visual brain responses as long as their set of natural language queries covers the relevant part of the representational space evoked by the objects in the images.

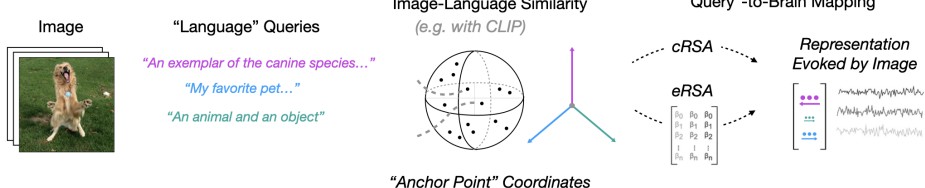

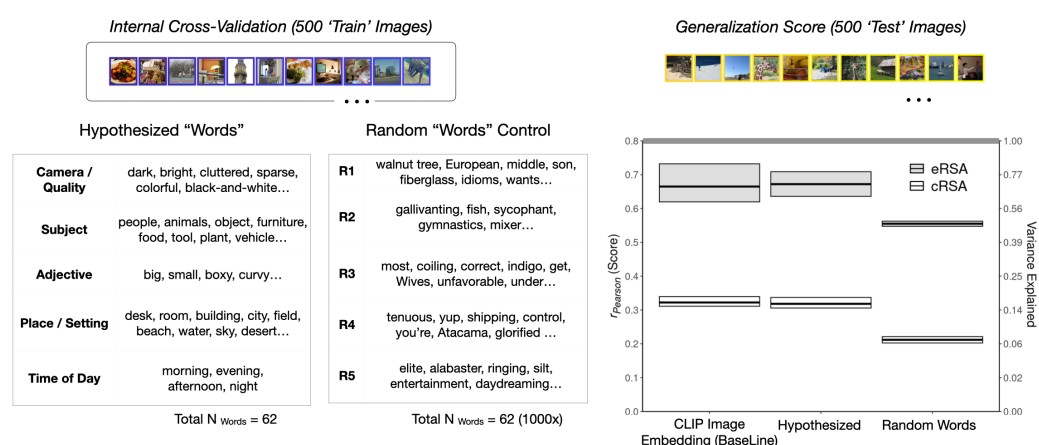

Figure 4: **Anchor Point "Word" Model Analysis**. In **A** we show a schematic of our analysis using arbitrary natural language queries embedded in CLIP to compute relative representations (Moschella et al., 2022), which are anchor points we subsequently use to predict image-evoked brain activity. This procedure involves first proposing candidate queries; then, computing the text embeddings for each of these queries; then, computing the text-to-image similarity of each image to each of the queries; and finally, using these text-to-image similarities as the basis of the mapping procedure to brains (cRSA or eRSA). In **B** we show the results of an experiment in which we contrast the predictivity of a 'hypothesis-driven' 62-Word model (derived from a mix of common words describing diverse object and scene attributes and CLIP-style prompt injections) with CLIP's image embeddings (768-D) and the mean of 1000 random 62-word selections. Note, that all 'development' of the anchor point word models is done via internal cross-validation on a training set of 500 images. The scores reported in the figure are the scores of the final models on a held-out test set of 500 images (and their associated captions).

## 3 DISCUSSION

In this work, we demonstrate that vision and language models predict visual cortex responses equally well, indicating that they may be converging on shared representational structure dominated by objects and agents (i.e. nouns). While significant future work will be necessary to further characterize this structure, our analyses suggest that this structure is obtainable either through purely visual bottom-up filtering operations over natural images, or, by performing multihead attention operations over the transition probabilities between words in the context of sentences. We further find both that attention and sentential context is largely unnecessary to match the predictive power of language in high-level vision. The 'or' of the first conclusion here is particularly important, as it argues against the idea that representations in high-level visual cortex are *predominantly* semantic and abstract in nature. To make this exclusion explicit, consider the evidence we would need for the opposite conclusion: a statistically significant case in which the unimodal language models or language-aligned vision models *outperformed* the unimodal vision models in prediction of OTC. Our sampling of the (recently SOTA) models from each of these categories does not provide such evidence.

We are not the first to demonstrate many of these trends. Long before the advent of sentence-processing language transformers, for example, Carlson et al. (2014) observed that the representational

geometry in occipitotemporal cortex is well captured by the semantic similarity features from earlier NLP models such as WordNet (Fellbaum, 2010). Similar observations were made by Huth et al. (2012), who used WordNet graphs to map a continuous semantic space of object and action categories that bridged visual and nonvisual cortex alike.

What does the surprising effectiveness of word-level language models in predicting high-level visual activity mean for the larger question of language alignment in the human visual system? Much of this answer depends on the weight we attribute to a number of limitations that otherwise scope or circumscribe our interpretation of the results we obtain from the particular models, metrics of representational alingment, and neural dataset that scopes our analysis.

**Limitations of the Data (ROI Subset)**: A first potential limitation is our particular treatment of data from high-level visual cortex in this study (deriving a target RDM from a single broad mask of occipitotemporal cortex). Popham et al. (2021) have previously suggested that language alignment in high-level visual cortex may be limited to a narrower subset of neural real estate in the most anterior portions of the ventral stream than those we have sampled here. Recent work from this same group has found that multimodal (language-aligned vision) models may be particularly adept at predicting the activity in these areas (Tang et al., 2023b). Others have analyzed data from the Natural Scenes Dataset at the voxel level, finding that models trained with language feedback can account for modest unique variance within certain brain areas such as the extrastriate body area (EBA) and some parts of the fusiform face area (FFA), relative an otherwise identical model that does not include language feedback (Wang et al., 2023).

**Limitations of the Data (Probe Stimuli)**: The Natural Scenes Dataset is arguably the current best dataset we have available for this kind of analysis in terms of reliability and stimulus-density – but it is far less optimal in terms of the particular stimuli it relies on. COCO images and their associated captions (solicited, by design, to be as visually grounded as possible) are almost certainly *not* the optimal stimuli for assessing the presence of linguistically interesting structure in high-level visual cortex. For this, we need far more targeted datasets evoking abstractions that language as a tool (c.f. Fedorenko et al., 2024) is particularly well-suited for.

**Limitations of the Models**: Perhaps overlooked in discussions about the limitations of the brain data and probe stimuli are the limitations of the candidate models themselves. The search for 'language-like' structure in the visual brain – through comparison to pure-language or language-aligned vision models – assumes such structure exists in the models (c.f. Doerig et al., 2022). However, an increasingly large body of empirical work challenges even this core assumption. For instance, CLIP and related models, including text-to-image systems like Stable Diffusion (Rombach et al., 2022), often lack basic compositional structure, such as distinguishing 'a spoon in a cup' from 'a cup on a spoon' (Conwell and Ullman, 2022; Yamada et al., 2023). These models frequently behave as 'bags-of-words' (Yuksekgonul et al., 2023), representing concepts in ways easily learned by simpler vision models. Even unimodal language models (LLMs) face similar criticism. Probes of entailment, negation, and counting suggest these models often fail to meet linguistic standards of finiteness, discreteness, syntactic well-formedness, and the ability to produce the new, but meaningful constructions that syntactic well-formedness allows (Thrush et al., 2022; Press et al., 2022; Bertolini et al., 2022; Hauser et al., 2002).

**(Cautiously) General Conclusions** The limitations above do potentially put somewhat strict and finite limits on broader conclusions we might make based on this work. Nevertheless, given the NSD's current centrality in the landscape of visual cognitive neuroscience datasets, and pending the emergence of datasets more deeply enriched with stimuli that elicit more complex linguistic structure, we do (for now) interpret our results as follows: Language alignment in the visual system is not an organizing force, but rather, a byproduct of the visual system's goal to capture statistically salient or ecologically relevant features of the visual world. Language models may align well with the high-level visual cortex because vision organizes the world into units that map easily to learned languages. These units, many of which correspond to distinct words, often have distinct visual features. In datasets like NSD, which comprehensively sample natural image statistics, these visual features account for most of the variance in visually evoked brain responses. Therefore, explicit language alignment may do little to improve predictions of the high-level visual cortex because language is already aligned with vision. Word-level models, even without sentence structure, appear to already capture the primary dimensions of this alignment.

Concurrent trends in machine learning also provide (preliminary) mechanistic evidence supporting this idea. Contrastive self-supervised learning models (Chen et al., 2020; Zbontar et al., 2021; Goyal et al., 2021a; Konkle and Alvarez, 2022), trained without any semantic supervision, readily learn features that support downstream object recognition with a single linear transformation. In language-alignment research, some have noted that algorithms such as CLIP (which backpropagates its alignment loss across the entirety of its vision and language encoders) may be inefficient because they do not leverage pre-existing structure in vision and language modalities: Alternatives to CLIP, such as LiT and DeCLIP, for example, maintain many of CLIP's advantages (e.g., zero-shot classification, robustness, guided conditional sampling) with largely frozen visual backbones pretrained via unimodal self-supervision (Li et al., 2022; Zhai et al., 2022). The success of these models suggests that whatever representational restructuring the language alignment task is doing, it need not be as deep or extensive as we think. Zooming out even further, the relative parity of vision and language models in predicting high-level visual cortical activity, may in some ways be a direct mirror of the datasets we've used to train machine vision models since the earliest days of deep learning. The canonical 1000 categories and 1.2 million images of ImageNet1K dataset (Deng et al., 2009) is a subset of a larger 14 million image dataset whose labels are 'synsets' from WordNet. From AlexNet onwards, then, our most popular visual models have often been trained on an image set whose primary dimensions of variance are defined by language.

This latter point underscores the profound challenge of disentangling vision from language and may explain why debates about which modality shapes the other are difficult to resolve conclusively (Konkle and Oliva, 2012; Grill-Spector and Weiner, 2014; Bracci and de Beeck, 2016; Long et al., 2018; de Beeck et al., 2023). Another trend we observe in the machine-learning literature is that large, well-trained models increasingly converge on similar representations, even without direct cross-modal learning (Pavlick, 2023; Huh et al., 2024). And while this convergence is illuminating, it highlights the need for more precise diagnostic tests to separate model representations in domains where they should be better specialized.

**Future Directions** Coming back to the question of dataset, then, we believe the data we really need is data that pushes perception to its natural limit, and therefore necessitates the involvement of language for understanding. Already, we are beginning to see work that uses the inherent 'abstractability' of language (i.e. more or less grounded descriptions of the same perceptual stimulus, predicated as well on factors like mutual familarity and active co-reference) to build stimulus sets specifically targeted at teasing apart vision from language. For example, recent work by (Shoham et al., 2024) using a custom dataset and iEEG recordings accords well with what we might expect based on the modeling we have done in this work: Language models given far more abstract descriptions of visual stimuli predict high-level visual brain data far worse than vision models given the visual stimuli directly. We consider this work an excellent step in the right direction, if not yet still the fullest picture we might obtain with similarly targeted datasets. 'Abstractability' need only be one tool in a diverse toolkit of probes that evoke the signature functional and representational structures we consider relatively more or even uniquely 'linguistic'. These structures include but are not limited to: compositional inversions and role-filler distinctions (i.e. the difference of transmitted meaning in 'man bites dog' versus 'dog bites man') (Frankland, 2015; Quilty-Dunn et al., 2023); external reference (i.e. the ability to represent 'cat' in the sentence 'the orange cat that I saw yesterday'); conceptual abstractions (e.g. justice); and (more generally) any kind of representational invariance that no combination of feed-forward visual filtering operations could feasibly produce (e.g. the representation of the written word 'apple' and a picture of an apple as the same) (c.f. Quiroga et al., 2005). Language tends to excel in domains requiring relational computations or logical operators, while vision excels in domains where granular or holistic distinctions (e.g., texture) are not easily expressible in natural language. Combining diagnostic tests that stress this difference with psychophysical approaches used by neuroscientists to probe for multimodality outside the visual cortex could help us determine when, where, and how truly "sentence-like" abstractions—those that can only exist through language—emerge in neural learning systems grounded in sensory perception.

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

# A  APPENDIX

## A.1  MATERIALS AND METHODS

### MODEL SELECTION

The models we test in this experiment consist of vision-only, language-only, and vision-language deep neural networks. These networks are supplemented with word-level models that operate over either the tokenized inputs of individual words (i.e. GLOVE) or directly over individual words (i.e. the count-vectorizing models).

Our sample of vision-only models consist of N=11 purely self-supervised visual contrastive-learning models from the VISSL model zoo (Goyal et al., 2021b). Our sample of hybrid vision-language models consists of N=10 models from the OpenAI CLIP, OpenCLIP, SLIP, and PyTorch-Image-Models repository (Radford et al., 2021; Ilharco et al., 2021; Mu et al., 2021; Wightman, 2019). Our sample of large language models (plus GLOVE) consists of N=12 models from Hugging Face and the SBERT repositories (Reimers and Gurevych, 2020; Wolf et al., 2019). To derive CLIP similarity scores for linguistic prompts, we use OpenAI CLIP's ResNet50 backbone (Radford et al., 2021). For part-of-speech extraction and word vectorizing operations, we use spaCy's English (BERT-Based) Large TRF model (Honnibal et al., 2020) and scikit-learn's CountVectorizer (Pedregosa et al., 2011), respectively.

### HUMAN FMRI DATA

The Natural Scenes Dataset (Allen et al., 2022) contains measurements of 73,000 unique stimuli from the Microsoft Common Objects in Context (COCO) dataset (Lin et al., 2014) at high resolution (7T field strength, 1.33s TR, $1.8mm^3$ voxel size). In this analysis, we focus on the brain responses to 1000 COCO stimuli that overlapped between subjects, and limit analyses to the 4 subjects (subjects 01, 02, 05, 07) for whom all 3 image repetitions are available for the overlapping images. The 3 image repetitions were averaged to yield the final voxel-level response values in response to each stimulus. All responses were estimated using a custom GLM toolbox ("GLMsingle" (Prince et al., 2022)), which was applied during the preprocessing of NSD time-series data, featuring optimized denoising and regularization procedures, to accurately measure changes in brain activity in response to each experimental stimulus.

**Voxel Selection Procedure**

To achieve a reasonable signal-to-noise ratio (SNR) in our target data, we implement a reliability-based voxel selection procedure (Tarhan and Konkle, 2020) to subselect voxels containing stable structure in their responses. Specifically, we use the NCSNR ("noise ceiling signal-to-noise ratio") metric computed for each voxel as part of the NSD metadata (Allen et al., 2022) for our reliability metric. In this analysis, we include only those voxels with NCSNR > 0.2.

After filtering voxels based on their NCSNR, we then filtered voxels based on region-of-interest (ROI). In our main analyses, we focus on voxels within the early visual and occipitotemporal cortices (EVC and OTC, respectively). For OTC, we first considered voxels within a liberal mask of the visual system ("nsdgeneral" ROI, see (Allen et al., 2022) for details). Next we selected the subset within either the mid-to-high ventral or mid-to-high lateral ROIs ("streams" ROIs). Then, we included all voxels from 11 category-selective ROIs (face, body, word, and scene ROIs, excluding RSC) with a t-contrast statistic ¿ 1; while many of these voxels were already contained in the streams ROIs, this ensures that these regions were included in the larger scale OTC sector. The number of OTC voxels included were 8,088 for subject 01, 7,528 for subject 02, 8,015 for subject 05, and 5,849 for subject 07, for a combined total of 29,480 voxels.

The EVC ROI encapsulates the ventral and dorsal aspects of areas V1, V2, and V3, as well as area hV4 (see (Allen et al., 2022) for details on ROI localization). To define the EVC ROI for each subject, we again first isolated voxels within the "nsdgeneral" ROI, and then selected for analyses any voxels that both fell within one of the early visual regions listed above, and that exceeded the NCSNR threshold of 0.2. This procedure yielded a total of 4,657 voxels for subject 01, 3,757 voxels for subject 02, 3,661 voxels for subject 05, and 3,251 voxels for subject 07.

**Noise Ceilings**

To contextualize model performance results, we estimated noise ceilings for each of target brain ROIs. These noise ceilings indicate the maximum possible performance that can be achieved given the level of measurement noise in the data. Importantly, our noise ceiling estimates refer to within-subject representational dissimilarity matrices (RDMs), where noise reflects trial-to-trial variability in a given subject. This stands in contrast to more conventional group-level representational dissimilarity matrices (Kriegeskorte et al., 2008a), where noise reflects variability across subjects. To estimate within-subject noise ceilings, we applied a novel method based on generative modeling of data's signal and noise characteristics (GSN; Kay et al., 2024).

This method estimates, for a given ROI, multivariate Gaussian distributions characterizing the signal and the noise under the assumption that observed responses can be characterized as sums of samples from the signal and noise distributions. A post-hoc scaling is then applied to the signal distribution such that the signal and noise distributions generate accurate matches to the empirically observed reliability of RDMs across independent splits of the experimental data. Noise ceilings are estimated using Monte Carlo simulations in which a noiseless RDM (generated from the estimated signal distribution) is correlated with RDMs constructed from noisy measurements (generated from the estimated signal *and* noise distributions).

FEATURE MAPPING METHODS

**Feature Extraction Procedure**

For each of our candidate DNN models, we extract features in response to each of our probe stimuli at each distinct layer of the network. At the end of our feature extraction procedure, for each model and each model layer, we arrive at a feature matrix of dimensionality number-of-images x number-of-flattened-features.

When using large language models, we obtain a single embedding from each model layer by averaging the 5 individual embeddings provided for each image. For both BERT-based and GPT-based models, we hook directly into the transformer layers to extract hidden representations. The tokenized captions are passed as a single tensor dataset, with token sequences padded to the maximum length. Attention masks are applied to ignore padding tokens. We apply no aggregation or pooling beyond that which is instantiated by submodules or functions in the feedforward pass. While non-standard, this approach allows us to apply a single, consistent across architectures, and was validated empirically with cross-reference to the embeddings obtained from standard Huggingface transfer-learning pipelines.

When using word-vectorizing (count) models (which do not contain layers), we obtain a single embedding by summing the word counts across the 5 image captions.

**Classical RSA (cRSA)**

To compute the classical representational similarity (cRSA) score (Kriegeskorte et al., 2008a) for a single layer, we used the following procedure: First, we split the 1000 images into two sets of 500 (a training set, and a testing set). Using the training set of images, we compute the representational similarity matrices (RSMs) of each model layer (500 x 500 x number-of-layers) using Pearson correlation distance metric. We then compare each layer's RSM to the brain RSM, also using Pearson similarity, and identify the layer with the highest correlation as the model's most brain-predictive layer. Finally, using the held-out test set of 500 images, we compute that target layer's RDM and correlate it with the brain RDM. This score serves as the overall cRSA score for the target model.

**Voxelwise Encoding RSA (eRSA)**

To arrive at a voxelwise encoding representational similarity (eRSA) score (Konkle and Alvarez, 2021; Kaniuth and Hebart, 2021) for a single model, the overall procedure was similar to that of cRSA, but with the addition of an intermediate encoding procedure wherein layerwise model features were fit to each voxel's response profile.

The first step in the encoding procedure is the dimensionality reduction of model feature maps. We perform this step for two reasons: (a) the features extracted from various deep neural networks can sometimes be massive (the first convolutional layer of VGG16, for example, yields a flattened feature matrix with 3.2 million dimensions per image); (b) the same dimensionality reduction procedure applied to all layers ensures that the explicit degrees of freedom across model layers is constant. To reduce dimensionality, we apply the scikit-learn implementation of sparse random projection

(Pedregosa et al., 2011). This procedure relies on the Johnson-Lindenstrauss (JL) lemma (Achlioptas, 2001), which takes in a target number of samples and an epsilon distortion parameter, and returns the number of random projections necessary to preserve the euclidean distance between any two points up to a factor of 1±epsilon. (Note that this is a general formula; no brain data enter into this calculation). In our case, with the number of samples set to 1000 (the total number of images) and an epsilon distortion of 0.1, the Johnson-Lindenstrauss procedure yields a target dimensionality of 5920 projections.

After computing this target dimensionality, we then proceed to compute the sparse random projection for each layer of our target DNN. The sparse random projection matrix consists of zeros and sparse ones of nearly orthogonal dimensions, and the layerwise feature maps are then projected onto this matrix by taking the dot product between them. The output of the procedure is a reduced layerwise feature space of size of 1000 images x 5920 dimensions with a preserved representational geometry. Note that in cases where the number of features is less than the number of projections suggested by the JL lemma, the original feature map is effectively upsampled through the random projection matrix, again yielding a matrix of 1000 x 5920 dimensions.

We compute our encoding model for each voxel as a weighted combination of these 5920 dimensions, using brain data from our training set of 500 images. (We note that while the number of dimensions needed for only 500 images would be only D=5326 according to the JL lemma, adding extra dimensions will only preserve the geometry with nominally less distortion than the epsilon provided, and does not affect the results). The fitting procedure for each voxel leverages SciKit-Learn's cross-validated ridge regression function, a hyperefficient regression method that uses generalized cross-validation to provide a LOOCV prediction per image (per output). This fit was computed over a logarithmic range of alpha penalty parameters ($1e^{-1}$ to $1e^{-7}$), to identify each voxel's optimal alpha parameter. We modified the RidgeCV function in order to select the best alpha using Pearson correlation as a score function (the same score function we use to evaluate the model at large), and to parallelize a slow loop for efficiency. This yielded a set of encoding weights for each voxel (number-of-voxels x 5920 reduced-feature-dimensions).

Next, with these encoding weights and the 500 training images, we compute the predicted response of every voxel to each image, and compute the corresponding *predicted* RSM using Pearson correlation. After computing each layer's RSA similarity value via Pearson correlation between the layer-predicted RDM and the target brain RSM, we again select the most predictive layer on the basis of results from the training set and compute this layer's RSA correspondence to the brain data using the held-out set of 500 test images. This test score from each model's most-predictive layer serves as the final eRSA score for each model.

We emphasize that this method contrasts with popular practices in primate and mouse benchmarking, which treat predictivity of unit-level univariate response profiles as the key measure. Because fMRI affords more systematic spatial sampling over the cortex, rather than taking the aggregate of single voxel fits as our key measure, we choose to treat the population representational geometry over each ROI as our critical target for prediction. This multi-voxel similarity structure provides different kinds of information about the format of population-level coding than do individual units (Kriegeskorte et al., 2008b). Computing the eRSA metric does, however, yield individual voxelwise encoding models, the individual predictive accuracies of which we register and have available in addition to the cRSA and eRSA scores for future analysis.

## A.2   SUPPLEMENTARY ANALYSES

Here, we report the results of two supplementary analyses in the comparison of pure-vision and pure-language models. In the first, we assess the OTC-predictivity of the randomly-initialized architectures corresponding to all our of unimodal (pure-vision or pure-language) models. In the second, we use a variance partitioning analysis (Lescroart et al., 2015; Lescroart, 2017) to assess the amount of brain-predictivity both shared between and unique to either modality.

Note that we have opted here to use the average voxelwise encoding scores that are implicit to our eRSA analysis, and report here scores only in the most-relevant contrast of occipitotemporal cortex (OTC). This is for multiple reasons: one) for simplicity; two) as a demonstration that the results derived from these scores (implicit to the eRSA analysis) largely concord with the RSA metrics used

in the main analysis; and three) for better synchrony with variance partitioning methods (which lend themselves naturally to regression-based metrics, but less so to RSA metrics).

## RANDOMLY-INITIALIZED MODELS

An important question in the comparison of pure-vision and pure-language models is the question of just how much we can attribute their divergent behaviors to differences in the modality of their training data alone. After all, pure-vision and pure-language models differ not only in the modality of their training data, but in the format of their inputs (tokenized strings versus pixels), their architectures (e.g. CNN versus ViT), and their training task (e.g. next-word prediction versus contrastive learning over various image augmentation regimes). While in our main analysis, we have attempted to try and abstract over these other differences by assessing a relatively diverse sample of each model type, another way we can probe the differences between them (at least at the level of architecture) is by running our same brain-prediction pipeline on the randomly-initialized versions of each.

In the case of this particular comparison between vision and language models, this comparison of trained versus randomly-initialized models has the added benefit of contributing directly to the logic of our interpretation. If, as we argue in the main body of the work, the 'language' in language-models does not yield altogether 'language-unique' or 'human language-like' representation (in the sense of having complicated structure that extends beyond the co-ocurrence statistics of common nouns, and not higher-order compositional meaning), then randomly-initialized versions of the language models should not suffer as substantial a decrease in performance without training as vision models will.

Indeed, we find this to be the case. In line with the idea that there is really not that much "language" structure in the language models, randomly-initialized language models only suffered a relatively minor drop in accuracy compared to their pretrained counterparts, with mean voxelwise encoding scores of $r_{Pearson}$ = 0.329 [0.327, 0.331] from trained weights and 0.276 [0.273, 0.279] for untrained (randomly-initialized) weights. We can contrast this with the far more substantive drop with randomly initialized vision models: with $rPearson$ = 0.341 [0.319, 0.367] for trained models and 0.156 [0.115, 0.193] for untrained models.

## VARIANCE PARTITIONING ANALYSIS

Variance partitioning (Lescroart et al., 2015; Lescroart, 2017) is an analysis technique that can be used to determine how much of the variance explained in a multivariate regression model is shared between or unique to the predictors. Here, we deploy this technique to partition the variance in OTC responses explained by our unimodal models – in this case, the most-brain-predictive models from each class (SEER-RegNet64GF for pure-vision; SBERT-MiniLM-L12-PML for pure-language). Variance partitioning in the case of our two predictors (pure-vision and pure-language) involves fitting a total of three regressions: two with each predictor alone, and one regression with the two predictors combined. In this case, the results of those regressions – in units both of $r_{Pearson}$ and $(r_{Pearson}^2)$ – are as follows:

1. OTC ~ Language (Alone): 0.354 (0.125)
2. OTC ~ Vision (Alone): 0.383 (0.147)
3. OTC ~ Language + Vision: 0.402 (0.161)

This produces the following unique and shared variances (in units of $r_{Pearson}^2$):

1. Unique to Language: 0.014
2. Unique to Vision: 0.036
3. Shared Language-Vision: 0.11

As we can see, the vast majority of the variance in brain prediction is shared between the pure-vision and pure-language models, though pure-vision has a slight advantage over pure-language in terms of unique variance.

