# OpenReview forum: "Rethinking Language-Alignment in Human Visual Cortex with Syntax Manipulation and Word Models"
_ICLR.cc/2025/Conference — Submitted to ICLR 2025_

### Official Review · Reviewer_wqdE · 2024-11-04

**Soundness:** 3
**Presentation:** 3
**Contribution:** 3
**Rating:** 6
**Confidence:** 3

**Summary:**

Inspired by recent successes of multimodal vision-language models in predicting visual cortical activity, the authors conduct a finer-grained analysis involving three sets of experiments on the Natural Scenes fMRI dataset. In the first experiment, the authors show that representations from vision-only self-supervised models are similarly predictive to representations from language-only models (trained via autoregressive and masked-language modeling losses). The second and third experiments probe how language-only representations can be predictive of occipitotemporal cortex activity, finding that token-based representations without explicit representation of syntax, be it from GLOVE embeddings or CLIP text embeddings for a base set of 62 hand crafted words, can achieve similar predictive results as normal language-based representations. From this, the authors argue that the similarity in predictive power of vision-only and language-only representations comes not from the influence of language on the high-level visual cortex but from them representing the same units in the world.

**Strengths:**

1) The experiments comparing GLOVE representations (with the whole sentence or a bag-of-words perturbation) against representations from language models make a convincing argument that at least for the Natural Scenes dataset, the variation can be explained by nouns and adjectives without requiring further syntactic structure. The following experiment with CLIP representation lends more support to this claim.

2) The authors present a strong discussion section highlighting various limitations of their methodology, most importantly the simplicity of the scenes in the Natural Scenes fMRI dataset not allowing for linguistically interesting captions that can be used to more convincingly validate the influence of language on the high-level visual cortex (or lack thereof).

3) The strength of the discussion makes it such that even if the results are not generalizable due to the limitations of the dataset, they point towards a limitation in the current datasets used in predicting brain representations from model representations and can catalyze future work curating datasets with richer scenes and captions.

**Weaknesses:**

1) While the authors note the chosen dataset and brain regions as limitations, they should also include the flaws of the model representations themselves as an obstacle in arriving at a clearer conclusion. CLIP representations have been shown to inadequately represent compositional structure, with CLIP language embeddings not being robust to changes in word order [1, 2] or minimal, meaning-altering edits to words [3, 4]. Given this, there is the alternative possibility that multi-modal representations do not improve over vision-only representations, not because the high-level visual cortex is not influenced by language, but because the representations do not adequately model additional syntactic structure.

[1] Thrush, Tristan, et al. "Winoground: Probing vision and language models for visio-linguistic compositionality." Proceedings of the IEEE/CVF Conference on Computer Vision and Pattern Recognition. 2022.
[2] Yuksekgonul, Mert, et al. "When and why vision-language models behave like bags-of-words, and what to do about it?." International Conference on Learning Representations. 2024.
[3] Ma, Zixian, et al. "Crepe: Can vision-language foundation models reason compositionally?." Proceedings of the IEEE/CVF Conference on Computer Vision and Pattern Recognition. 2023.
[4] Hsieh, Cheng-Yu, et al. "Sugarcrepe: Fixing hackable benchmarks for vision-language compositionality." Advances in neural information processing systems 36 (2024).

**Questions:**

The feature extraction procedure section in the appendix would benefit from more details. For instance, for BERT-based models, is the CLS token used as is convention or is some separate aggregation procedure applied on hidden representations associated with each token? Likewise for GPT-2, is the embedding of the final token used as a representation of the caption?

---

> ### Comment · Reviewer_wqdE · 2024-12-02
> **Response to authors**
>
> I thank the authors for addressing the shortcomings of existing vision-language model embeddings in their writing. I will be maintaining my score as the shortcomings of models at representing language structure still affects what inferences can be drawn about representations in the high-level visual cortex.
>
> At the same time, however, I want to note that the added paragraph strengthens an already robust and nuanced discussion section that places the results in context and surveys valuable directions for future work.

---

> > ### Author Response · Authors · 2024-12-04
> > **Primary Response to Reviewer wqdE**
> >
> > We thank the reviewer for their feedback and their engagement with our work, and apologize for the delay in our point-by-point response. We appreciate that the reviewer has already taken into account our revision, even in the absence of a personalized response!
> >
> > In the interests of discussion (if the reviewer has time) we wanted to clarify one point regarding the “limitations of the models themselves”. Is the reviewer’s concern here related only to the multimodal (e.g. CLIP-like models) or does the concern extend as well (or to a similar degree) to the “pure-language” models (e.g. BERT, GPT2)? If the former (multimodal vision models), we agree with the reviewer that the outlook still appeals largely to be bleak -- though we could try models we’ve recently discovered in the process of this revision (e.g. the NegCLIP we have already included in the revision in response to another reviewer’s suggestion). Structure-CLIP [1], for example, is a model that augments the CLIP training paradigm with structured scene graphs. In terms of “pure-language” models, we similarly do not have high hopes in any modern neural language model to satisfyingly account for compositional syntax, but there are models we might consider (e.g. state-space and hybrid models such as Taipan, Samba, or SPADE) that are supposedly more syntax-aware in their modeling of long-range dependencies. Again, we are not sure there’s much gain to be made here, nor much conceptual understanding in the absence of more targeted brain datasets, for example, but…
> >
> > Were we to assess these models (or other models the reviewer might have in mind), would the reviewer consider raising their score?
> >
> > [1] Huang, Y., Tang, J., Chen, Z., Zhang, R., Zhang, X., Chen, W., ... & Zhang, W. (2024, March). Structure-CLIP: Towards Scene Graph Knowledge to Enhance Multi-Modal Structured Representations. In Proceedings of the AAAI Conference on Artificial Intelligence (Vol. 38, No. 3, pp. 2417-2425).

---

### Official Review · Reviewer_NGEG · 2024-11-04

**Soundness:** 3
**Presentation:** 3
**Contribution:** 3
**Rating:** 6
**Confidence:** 2

**Summary:**

This paper surveyed whether high-level visual representations in the human brain are aligned with language by investigating how well the language-only model can predict neural responses to images compared with that of a vision-only model. Using the Natural Scenes fMRI Dataset, they find that while language models predict brain responses and vision models, the predictive power of language models reflects their ability to capture information about nouns present in image descriptions rather than syntactic structure or semantic compositionality. The authors imply that such convergence between the language and vision representations in the high-level visual cortex is due to a common reference to real-world entities rather than by direct interaction of vision and language.

**Strengths:**

- Novel analysis with "anchor point embeddings" for illustrating simple word-based predictions
- Comprehensive experimental design comparing multiple model types (vision-only, language-only, hybrid)
- Good empirical evidence supporting their main claims

**Weaknesses:**

- Without testing on more diverse datasets (apart from NSD), how are we making sure that there's no bias in the analysis
- Captions, like those in the COCO dataset, are designed to describe static scenes and thus yield a dataset that is inherently biased toward simple object-focused descriptions. This kind of language may correspond well with visual cortex responses to simple image recognition tasks but does not allow for an understanding of how the brain integrates language in more complex or conceptual ways
- RSA can overgeneralize the degree of alignment, masking differences that might appear in a voxel-wise or layer-specific type of analysis.

**Questions:**

- What are the author's thoughts on extending the analysis to include dynamic stimuli or temporal sequences?
- How do you you think the results might/might not change with more abstract or conceptual visual stimuli that require linguistic knowledge for interpretation?

---

> ### Author Response · Authors · 2024-12-04
> **Primary Response to Reviewer NGEG (Part 1)**
>
> We thank the reviewer for their feedback and engagement with our work. Please find our point-by-point responses to your questions and concerns below:
>
> Weaknesses:
>
> - “more diverse datasets” / “bias in the analysis”: As much as it pains us, we do ultimately agree with the reviewer that the sole and primary dataset we use in this analysis is one of the study’s more substantial weaknesses. We take care to note this (extensively) in the discussion, but also again want clarify that our use of the Natural Scenes Dataset is both a deliberate and an arguably necessary default. Multiple publications in the past few years have made claims about language-alignment on the basis of model comparisons using only this dataset [1, 2], and one of our primary intents with this work was to show (using more or less the same paradigm as many of these other works) that what has previously been proposed as evidence of ‘language-alignment’ in the visual cortical activity measured by the NSD may not be as linguistic as it initially seems. That being said, we are fully in accord that if we want to make this claim *more comprehensively* and *generalizably*, we very much should expand the paradigm of vision / language model comparison to other datasets. As we expand on more in a point below, a recent study (published since the submission of this work) has done exactly this, with what we consider to be very incisive implications about the possibility of language-alignment in the visual brain [3].
> - “Generalizability from COCO captions”: In short, we agree word for word with the reviewer on this front, and believe firmly this is one of the primary reasons why we see little difference between the vision and language models in this analysis. More on this in our discussion below of a recent paper [1] that uses more and less grounded “captions” and a targeted neurophysiology (iEEG) dataset to address this directly.
> - “RSA can overgeneralize / mask differences”: While we generally agree with key parts of the reviewer’s concern here, there are a few caveats and further considerations we should take into account:
>   - Spatial specificity in the brain: We agree with the reviewer that our use of a broad mask of high-level visual cortex (OTC, between ~5800 and ~8000 voxels per subject after functional and reliability-based selection) may be masking subtler differences between smaller sub-parts of high-level visual cortex (including e.g. “category-selective” ROIs such as FFA and PPA) or even voxel-specific differences. Our claim is thus very much circumscribed to a notion of  “high-level visual cortex” as a computational monolith, with higher-order properties that may yield different representations at different locations on the topographic cortical map it encompasses, but which may ultimately be characterized by a finite set of broad, overarching computational principles (hierarchy, separability, invariance, consistency). Our primary conclusion, therefore is just that  ‘alignment to language’ does not yet appear to be one of these broad, overarching principles (at least inasmuch as our current best-in-class, largest, most reliable and most comprehensive neuroimaging dataset (i.e. the NSD) is concerned).
>     - All this being said, there may be more to the story, and as we mention in the original and revised Discussion (lines 442-451, now explicitly tagged), there does continue to be some debate about more specific ROIs and voxel-level differences that we do not arbitrate on in this work.
>   - Specificity of the modeling procedure: While we do not report sub-ROI or voxel-level differences, one important consideration we do want to highlight is that that we are using two methods of RSA: unweighted, “classical” RSA (cRSA) and “voxel-encoding” RSA (eRSA). We agree with the reviewer that the first of these (cRSA) applied to a large-scale neural sector is a relatively coarse method for studying alignment. Importantly, though, our second RSA procedure (eRSA) approach is actually both a voxel-wise and layer-specific analysis procedure. eRSA involves guided re-weighting of the DNN features [4, 5], under the assumption that different voxels are likely tuned to different features, and thus, that each voxel’s response profile should be modeled as a weighted combination of the units in a layer, using independent brain data for fitting the encoding model. After fitting, each voxel-wise encoding model is used to predict responses to the test images, for which we compare the predicted population geometry to the observed population geometry of neural responses. While this last step involves comparing similarity matrices, just as in classical RSA, it is critical to note that the model-derived RDM is generated from a set of voxel-wise encoding models. Thus, the encoding procedure should be flexible enough to reveal any differences of the sort the reviewer mentions.

---

> > ### Author Response · Authors · 2024-12-04
> > **Primary Response to Reviewer NGEG (Part 2)**
> >
> > (Continuing from Part 1...)
> >
> > Weaknesses (continued)
> > - Specificity of the layer-wise selection: A final (briefer point) just to say that we again strongly agree with the reviewer that the assumption and comparison of models on the basis of single “layer-wise” linear correspondences likely obscures finer details and differences between models that hierarchical, multi-layer, or non-linear correspondences might be more sensitive to. This is currently a very active area of research in the field of NeuroAI! but without clear consensus yet as to how we should move beyond the current defaults. Without this consensus, we decided to stick with those defaults -- familiar as they are to our target audience, and commensurate with those works that are our most direct empirical foil.
> >
> > Questions
> >
> > - “extensions to dynamic stimuli or temporal stimuli sequences”: We agree wholeheartedly with the reviewer’s intuition this is an excellent direction for future versions of this analysis or similar lines of work. The meaning of linguistic utterances -- especially with respect to grounded or external referents -- seems inherently to unfurl (and change!) over time, and there are a wide variety of well-documented psychophysical phenomena (e.g. the garden path effect, surprisal writ large) and neurological phenomena (e.g. the N400 and P600 ERP components) related to this temporal processing. In the context of visuo-semantics, tasks requiring relational reasoning and affordance estimation (which could be linked to visual search and saccadic scanpath measurement) seem to us particularly relevant targets for the collection of new behavioral and neuroimaging datasets. In short, tasks that push the envelope of the dynamic “resolution” that seems to be a cornerstone of the vision-language hand-off are in our minds precisely the kinds of tasks that could help to better differentiate both the idiosyncratic contributions of vision and language to any given behavior / neural response profile, and the underlying vision and language models we hypothesize to be some approximation of the computations involved. (A major asterisk here to note, as many of the reviewers have mentioned, is that it remains largely unclear whether this approximation is valid in the first place.) Already we see in some analyses that look for multimodality in brain recordings that temporality seems to play a nontrivial role in the difference between vision, language, and language-aligned vision models [6].

---

> ### Author Response · Authors · 2024-12-04
> **Primary Response to Reviewer NGEG (Part 3)**
>
> (Continuing from Parts 1 and 2)
>
> Questions (Continued)
>
> - “change with more abstract / conceptual stimuli”: One of our primary motivations in pursuing this work was a default operating hypothesis that the prediction of visual brain activity by pure-language and language-aligned vision models (observed concurrently by multiple research groups) had very little to do with language, and far more to do with the particular set of stimuli used when measuring the visual brain activity. By extension, we also assumed that this ‘alignment’ would not extend to certain kinds of brain activity that seemed a priori to involve operations well beyond the purview of either modality (vision or language) alone. (One example we would often discuss as a group were ‘gnostic neurons’ in hippocampal regions, i.e. the ‘Jennifer Aniston’ cell, with representational invariances that spanned images, text, and audio alike -- an invariance we thought could not feasibly have emerged from purely feedforward visual operations. Another example were what we would often refer to as ‘ungrounded’ concepts like ‘love’ or ‘justice’).
>   - Perhaps needless to say, then, it has long been our belief that measuring the representational alignment between brain activity and vision / language models given more “abstract / conceptual” stimuli would produce a very different set of results than the ones we obtain in this particular analysis with this particular brain data (measured over COCO images and captions). And as of just two weeks ago, we may now actually have some direct empirical backing on this front thanks to the aforementioned paper [1] (a preprint published after our original submission) that contrasts the representational alignment of vision and language models on a dataset explicitly designed to elicit different levels of conceptual abstraction. In line with both our default hypothesis and our empirical results in the current work, this work demonstrates that vision and language models yield similar predictions only when they are compared on the basis of captions that refer more concretely to observable properties of the image. When they are compared on the basis of more “abstract” captions (e.g. the Wikipedia article associated with a given image), they begin to look substantially different -- with language models, especially, losing much of their alignment to the image-evoked visual brain activity. This is precisely the kind of work we hoped to inspire with our current submission, and are immensely pleased to see the field moving in this direction, and we have now mentioned / cited explicitly in our revised discussion (last paragraph, lines 511-533).
>
> [1] Wang, A. Y., Kay, K., Naselaris, T., Tarr, M. J., & Wehbe, L. (2023). Better models of human high-level visual cortex emerge from natural language supervision with a large and diverse dataset. Nature Machine Intelligence, 5(12), 1415-1426.
>
> [2] Doerig, A., Kietzmann, T. C., Allen, E., Wu, Y., Naselaris, T., Kay, K., & Charest, I. (2022). Semantic scene descriptions as an objective of human vision (arXiv: 2209.11737). arXiv.
>
> [3] “The organization of high-level visual cortex is aligned with visual rather than abstract linguistic information”, Shoham et al, BioRxiv, 2024
>
> [4] Kaniuth, P., & Hebart, M. N. (2022). Feature-reweighted representational similarity analysis: A method for improving the fit between computational models, brains, and behavior. NeuroImage, 257, 119294.
>
> [5] Konkle, T., & Alvarez, G. A. (2022). A self-supervised domain-general learning framework for human ventral stream representation. Nature communications, 13(1), 491.
>
> [6] Subramaniam, V., Conwell, C., Wang, C., Kreiman, G., Katz, B., Cases, I., & Barbu, A. (2024). Revealing vision-language integration in the brain using multimodal networks. In International conference on machine learning. PMLR.

---

### Official Review · Reviewer_ohqL · 2024-11-04

**Soundness:** 2
**Presentation:** 3
**Contribution:** 2
**Rating:** 5
**Confidence:** 4

**Summary:**

This paper looks at the alignment between visual cortex activity and vision/language models, and investigate the idea that high-level visual representations are language-aligned. Using fMRI data from the Natural Scenes Dataset (NSD -- specifically a 1000 images subset), they compare the ability of vision-only, language-only, and multimodal (vision-language) models to predict brain responses in early visual cortex (EVC) and occipitotemporal cortex (OTC).

They find that unimodal language models predicted OTC activity as well as unimodal vision models, but this predictive power stemmed primarily from capturing information about nouns in image captions, rather than syntactic structure or semantic compositionality. A simplified “handcrafted” word model based on just 62 nouns and adjectives, using CLIP embeddings, performed comparably to the full CLIP image embeddings in predicting OTC activity. This suggests that the success of language models in predicting high-level visual cortex activity may be due to capturing grounded information from co-occurrence statistics rather than true language alignment.

Language models performed significantly worse than vision models in predicting EVC activity, highlighting their inability to capture lower-level visual features.

**Strengths:**

Broadly I do think the authors did a good job designing the experiments, and the problem is of concrete scientific interest (rather than of only engineering interest like many image decoding works that rely on NSD). The authors go to a great extent in ensuring the soundness of their experiments.

The question that authors pose in the paper (to what extent do linguistic features of image descriptors predict OTC activations) is very interesting, especially in context of prior work that has been suggestive of linguistic integration at the edge of what are traditional "visual areas" in the brain.

**Weaknesses:**

I have two primary concerns:
* The first is regarding the breadth and scope of the claims, and if the current experiments are really sufficient to substantiate the claims.
* The second is regarding the third experiment which utilized CLIP

Regarding the former -- "scope of claims". For example the abstract ends with `prediction of brain data whose principal variance is defined by common objects in common, non-compositional contexts` and the introduction from `Line 046` to `Line 067`. My primary concern is that it is unclear if:
1. Perhaps it is a limitation of current language models, and they simply do not capture spatial relationships or complex (negation or adjective) relationships. Is it not possible, that one day there may exist some language-only model that captures these compositional relationships well? [1, 2, 3]
2. It is unclear if the claims are due to limitations in the fMRI modality, which primarily reflects slow temporal signals. Would the claims necessarily still hold up under electrophysiology or calcium imaging? I think this is at least worth discussing.

[1] Locating and Editing Factual Associations in GPT (specifically this paper identifies auto-regressive language models as having directional fact attributions, which clearly breaks the compositionality assumption)

[2] Evaluating Spatial Understanding of Large Language Models

[3] What's "up" with vision-language models? Investigating their struggle with spatial reasoning

Regarding the latter concern (third experiment):
* It is well known that the text encoder from CLIP and other vision-language contrastive models behave like a bag-of-words model and rarely perform above chance on compositional or spatial reasoning tasks, and this is likely due to the dataset used to train these models [4, 5, 6, 7, 8]. These issues are widely known, and it is concerning to me that the authors were seemingly unaware of this issue. I believe this experiment is primarily showing this failure of current vision-language models, and it is difficult to use this experiment to make claims about representations in the brain.

[4] When and Why Vision-Language Models Behave like Bags-Of-Words, and What to Do About It?

[5] Winoground: Probing Vision and Language Models for Visio-Linguistic Compositionality

[6] Breaking Common Sense: WHOOPS! A Vision-and-Language Benchmark of Synthetic and Compositional Images

[7] Why is winoground hard? investigating failures in visuolinguistic compositionality

[8] When are Lemons Purple? The Concept Association Bias of Vision-Language Models

Another minor concern is the extent of the machine learning contribution. While this paper was submitted under the primary area "applications to neuroscience & cognitive science", I do not think this paper makes any claims regarding new machine learning techniques. However I do want to emphasize that this is a minor concern.

Other very minor issues:
1. Text format of the paper does not match other ICLR papers (font choice/thickness, spacing)
2. Wrong paper template (Does not show "Under review as a conference paper at ICLR 2025")

**Questions:**

1. Why were 1,000 images chosen? Was this because these 1,000 images were seen by all four subjects?
2. Table 1, what is the model used in "Vision-only" or "Language-only"? Is this one model? Or is this the average of multiple models?
3. For Line 151, is that the noise ceiling averaged over all OTC voxels?
4. What token are you using for the next token or masked language models as input to the encoder?
5. For Figure 2, when you discuss multimodal vision models, are you using the image component or the text component?
6. For Figure 2, can you clarify what is the "semiopaque fill" and "translucent fill"? To me they look the same.
7. How are the count models used? Do you have every potential word initialized to 0, and then set the corresponding lookup to the number of occurrences for each word?

---

> ### Author Response · Authors · 2024-12-02
> **Primary Response to Reviewer ohQL (Part 1)**
>
> We thank the reviewer for the thoughtful review and constructive feedback, as well as the opportunity to clarify and improve our work. Below, we address the reviewer’s questions and concerns point by point.
>
> (Please note, a last-minute versioning issue meant that some of the modifications we intended to include based on the reviewer’s feedback do not appear in the most recent revision. Please accept our apologies for this error. We will be sure to correct it in the final version of the manuscript should it be accepted).
>
> Primary Concern 1: Scope / Breadth of Claims
>
> - *On limitations of current language models*: The reviewer raises an important point about whether future language-only models might better capture spatial relationships or compositional semantics. We agree that this is a possibility and have **clarified in the manuscript that our findings are constrained to the particular set of models we use in this analysis (lines 436-440)**, however representative they are of the current state-of-the-art with respect to LLM syntax comprehension. Specifically, we do not claim that language-only models cannot one day achieve such capabilities, but rather that their current performance in predicting neural activity appears limited to grounded object-level representations rather than compositional semantics.
>   - We have also cited the relevant work (\[1\], \[2\], \[3\]) that aligns with your observation about the limitations of current language models. **These have been incorporated into our revised Discussion section (lines 440-471)** to acknowledge this broader context.
>
> - *On fMRI modality limitations*: Yet another point you raise is on the limitations of fMRI as a measurement modality and whether our claims would generalize to techniques with finer temporal resolution, such as electrophysiology or calcium imaging. While space constraints did not allow us to expand the discussion significantly beyond the main revision (updated to discuss the limitations of the language models), we initially added a new appendix (“Other Limitations of the Dataset”) with a targeted subsection to address this point directly (though  please see the “versioning issue” note above).
>   - "Another limitation of the fMRI dataset in our study (and of fMRI in general) is temporal resolution. fMRI is deeply constrained and slow in its ability to capture temporal signal and may not reflect transient neural dynamics that could encode more sophisticated compositional semantics. Future studies using methods such as electrophysiology or calcium imaging, which provide finer temporal resolution, could test whether such representations emerge under different temporal scales or experimental paradigms.”
> - While our findings suggest limited alignment at the timescale measured by fMRI, we acknowledge the importance of investigating this question using complementary methods in future work. At the same time, we should in due diligence acknowledge that fMRI has particular advantages for studying phenomena relating to visual-language alignment in humans – the alignment between visual and linguistic representations necessitates examining distributed neural activity across broad swaths of visual and associative cortex—a level of coverage that intracranial electrodes rarely achieve. Adding to the appendix above, we address this proviso as follows:
>   - "At the same time, fMRI is well-suited for this study due to its ability to provide comprehensive spatial coverage of the entire human brain, including the high-level visual and occipitotemporal cortices critical for investigating visual-language alignment. While intracranial recordings offer superior temporal resolution and direct neural measurements, their use is inherently limited by the clinical constraints of electrode placement, which often precludes comprehensive coverage of the regions most relevant to this research. Specifically, the alignment between visual and linguistic representations necessitates examining distributed neural activity across broad swaths of visual and associative cortex—a level of coverage that intracranial recordings rarely achieve. Furthermore, fMRI allows for data collection in healthy individuals rather than restricting the study to patients undergoing invasive monitoring, thereby enabling a broader and more representative investigation of these neural processes."

---

> > ### Author Response · Authors · 2024-12-02
> > **Primary Response to Reviewer ohQL (Part 2)**
> >
> > (....Continuing from the previous comment: Part 1)
> >
> > Primary Concern 2: Experiment 3 and CLIP’s Bag-of-Words Behavior
> >
> > We appreciate your observation about CLIP’s limitations in capturing compositional and spatial reasoning, as documented in recent work ([4], [5], [6], [7], [8]). We have now included citations to these studies in our revised discussion (lines 440-471), noting explicitly that CLIP-like models have been shown to behave similarly to a bag-of-words model. Crucially, we believe strongly that this known limitation directly aligns with our findings and supports the idea that the predictive power of language models is rooted in their ability to capture object-level co-occurrence statistics.
> >
> > To clarify the logic of Experiment 3 as it pertains to this point: Our use of the phrase “handcrafted word models” is based directly on our intuition (shared it seems by most of the reviewers) that effectively all of the alignment to neural activity offered by the language models can be reduced to a custom “bag of words” -- so long as this “bag of words” instantiates a relative representation with sufficient coverage of the underlying manifold shared between the models and the brain (with no more than a linear transformation in between). Our “handcrafted words” are quite literally a “bag of words” -- and both the hypothesized word set and the random word set are designed explicitly to exclude compositional constructions.
> >
> > In short, Experiment 3 serves effectively as a demonstration / conceptual replication of the “bag-of-words” behavior observed in other uses of CLIP-like (multimodal, language-aligned) vision models. We hope this clarification sufficiently assuages the reviewer’s concerns.
> >
> > Machine Learning Contribution
> >
> > We agree with your observation that our paper does not propose new machine learning methods. However, as you note, our primary focus is on neuroscience and cognitive science applications. We initially inserted the following clarification to our introduction (and will again if the manuscript is accepted or we are able to resolve the versioning issue noted above).
> > "We emphasize that the primary contribution of this work is not in introducing new machine learning methods but in leveraging and rigorously applying existing computational tools to advance our understanding of neural representations. By systematically evaluating the alignment between visual cortex activity and representations derived from an array of language models, we can develop better understanding of the pressures guiding representation formation in the brain."
> >
> > Addressing “Minor Concerns”
> > - Formatting issues: We made sure that the updated manuscript used the most-up-date version of the ICLR-2025 LaTeX template and style files. (We note that the 2025 template does not appear to include the footer mentioned by the reviewer. Perhaps this is a recent change to the template?)
> > - Clarifications in figures and tables:
> >   - Table 1: We now clarify that "Vision-only" and "Language-only" are the average scores across all models in their respective categories directly in the table.
> >   - Figure 2: The "semiopaque" and "translucent" fills have been replaced with more intuitive terminology and indicators (“eRSA: ‘Filled Cross-Bars, Higher Scores’; cRSA: ‘Hollow Cross-Bars, Lower Scores”).
> >   - Count models: Word counts are indeed initialized to 0, with values updated based on the occurrence of each word. (We initially added this information to the “Feature Extraction Procedure” subsection of the Methods Appendix, and will include it moving forward in resolution of the versioning issue noted above.)
> >
> > Responses to Specific Questions
> >
> > - "Why 1,000 images?": Yes, we chose 1,000 images because these were seen by all four subjects, allowing for consistent analysis across participants. This is mentioned in the “Human fMRI Data” Appendix (lines 778-786).
> > - Noise ceiling (Line 151): Yes, this is the noise ceiling averaged over all OTC voxels. (We initially added this information directly to the caption of Figure 2, and will again in resolving the versioning issue noted above).
> > - Tokens used for language models: We use sentence-level inputs, tokenized with each model’s native tokenizer. This has been clarified in the Methods section (lines 834-841).
> > - Multimodal models in Figure 2: For multimodal models, we use only the vision encoders. (We have considered using the text encoders in the past, though ultimately chose not include them, since they did not seem particularly relevant to the theoretical question of `language-alignment’ as it impacts visual representation and are not often used after training in machine-learning applications of multimodal models).

---

> ### Comment · Reviewer_ohqL · 2024-12-02
>
> Thank you for your revision. They do address some of my concerns. I have raised my score to a 5.
>
> There are two reasons I hesitate giving a higher score:
> 1.  **My main concern lies in the scope of the claims and if the discussion (both here in the rebuttal and the revision) are sufficient to clarify this point.** I don't think my concerns around this point have been fully addressed, but I appreciate that the authors have thought about this and have made an serious attempt at addressing this.
> 2. I think the machine learning contribution is very limited (if any).

---

### Official Review · Reviewer_pTbq · 2024-11-04

**Soundness:** 3
**Presentation:** 4
**Contribution:** 2
**Rating:** 6
**Confidence:** 3

**Summary:**

The paper studies how language-only representations of image captions can predict image-evoked human visual cortical responses, both occipitotemporal cortex (OTC) and early visual cortex (EVC). The paper includes three studies that argue that pure visual learning and pure language learning may be converging on representations that are equally predictive and that the nouns account for the performance of language models; specifically, they act like bag-of-word models.

**Strengths:**

The paper is well-structured and clearly written. The methodology, results, and discussions are interesting and easy to follow. The authors not only provide empirical findings on the effectiveness of the language and vision representations but also take a closer look at what actually contributes to the observations.

**Weaknesses:**

See Questions

**Questions:**

- I wonder why the authors choose only one model (SBERT-MiniLM-6) as the only representation of the LLMs in the experiment in Section 2.2 while the results from Figure 2 and the discussion in Section 2.1 argue that this is the standout, which could be an outlier and doesn't fully represent the language-only model. Have the authors seen similar results when experimenting with different LLMs?
- There is a discussion on how a vision-language model like CLIP performs like a bag of words [1, 2, 3]. If I understand correctly, not enforcing differentiation between the correct caption and shuffled caption can cause a loss of the capability of compositional reasoning, which the authors discussed in Section 3. It would be great if you could experiment by substituting with improved models like NegCLIP in [1] for text embedding.

[1] When and why vision-language models behave like bags-of-words, and what to do about it? Yuksekgonul et al., ICLR 2022
[2] SUGARCREPE: Fixing Hackable Benchmarks for Vision-Language Compositionality, Hsieh et al., NeurIPS 2023
[3] TripletCLIP : Improving Compositional Reasoning of CLIP via Synthetic Vision-Language Negatives, Patel et al., NeurIPS 2024

---

> ### Author Response · Authors · 2024-12-04
> **Primary Response to Reviewer pTbq**
>
> We thank the reviewer for their positive feedback, and appreciate the engagement. We want to begin by highlighting the main change(s) made to the manuscript as a direct function of the reviewer’s feedback: In this case, **the inclusion of NegCLIP in our set of multimodal model candidates (see updated Figure 2\) and a substantively revised Discussion section discussing the “bag of words” phenomenon in CLIP-like models (lines 459-471)**. NegCLIP’s vision encoder performs at around the same level as other multimodal vision encoders in both cRSA and eRSA.
>
> As per NegCLIP’s text encoder, we were somewhat on the fence with how to proceed, as we generally did not include multimodal text encoders in our suite of model candidates, given that these are not often used for inference or for embedding-based transfer-learning after training. (Even captioning models that use multimodal embeddings tend to use just the vision encoder, and then an independent causal language model with a token adapter for the caption generation. Similarly, and at the same time, many recent CLIP-like models now use frozen text encoders with adapters in lieu of end-to-end gradient training of text and vision simultaneously.) That being said, if results for the multimodal text encoders are of interest, we are happy to report them here, and then to include them as part of our supplementary materials if the reviewer considers them relevant.
>
> (Please note that with respect to the response below, a last-meant versioning issue resulted in some of these updates not appearing in the most recent version of our manuscript. We apologize for this error, but fully intend to include these updates in the final version if the manuscript is accepted. We thank the reviewer in advance for their understanding\!)
>
> *On the language perturbation experiment (Section 2.2) with models other than SBERT-Mini-LM6*: The reviewer does make a good point that we simultaneously highlight SBERT-Mini-LM6 as somewhat of a standout, but then use it as our “representative” LLM contrast in the language perturbation experiments. In retrospect, we realize this might be somewhat confusing. This is in part because the model is representative in the sense that the average of its cRSA and eRSA scores makes it one of the closest models to the overall LLM mean across experiments \-- but we realize now this is because its cRSA score is far above average and its eRSA score is slightly below average. We have updated this section (with an update lost in the versioning issue mentioned above, but which we very much intend to rectify) now to include an additional model (SBERT-MPNet, close to the average LLM in both cRSA and eRSA) to demonstrate that results hold in other LLMs as well. We provide a side-by-side comparison of the original MiniLM-L6 and the new MPNet results below:
>
>   Region | ModelVariant                             | MeanScore [LowerCI, UpperCI]
>
> OTC  |   Bag of Nouns (SBERT MPNet)     |    0.441   [0.304,   0.576]
>
> OTC  |  Bag of Nouns (SBERT MiniLM-L6)   |     0.415  [0.296,  0.530]
>
> OTC  |  Bag of Verbs (SBERT MPNet)       |   0.323   [0.207,   0.442]
>
> OTC  |  Bag of Verbs (SBERT MiniLM-L6)    |     0.309   [0.232,  0.386]
>
> OTC  |  Shuffled Words (SBERT MPNet)  |    0.485   [0.396,   0.612]
>
> OTC  |  Shuffled Words (SBERT MiniLM-L6) |    0.463   [0.347,   0.580]
>
> OTC  |  Original (SBERT MPNet)              |    0.526   [0.422,   0.636]
>
> OTC  |  Original (SBERT MiniLM-L6)        |         0.546   [0.465,   0.627]
>
> In short, the results with MPNet on the word perturbation experiments are largely commensurate with those of the MiniLM-L6 \-- (and statistically, the difference of differences between the two across different perturbations is not significant). All the key findings that tell us both how little word order (and words that are not nouns) matter hold just as strongly (if not more strongly) with the MPNet variant. The key revision we intend(ed) to make to the manuscript, then, was simply to update the description of Experiment 2.2 (currently lines 266-268), to read:
>
> - “We test these degradations in 4 models: SBERT-MiniLM-6 (the standout model in CRSA) and SBERT-MPNet (one the LLMs closest to the LLM average in both cRSA and eRSA), which both learn over sentences; count vectors, which involve no learning at all; and GLOVE models, which learn only over individual words and without hierarchical attention operations.”

---

### Author Response · Authors · 2024-12-04
**Global Response (following Point-by-Point Responses)**

Dear Reviewers,

Thank you to all for your engagement with our work, and please accept my apologies as lead author for the tardiness / patchiness in posting them. Despite some setbacks and a versioning issue with the updated manuscript (which I have noted in the relevant point-by-point responses), I hope we have now addressed all of the initial feedback you have so kindly taken the time to give.

I will be doing my best to respond as promptly as possible to ongoing discussion in the days left before discussion closes, and (though the manuscript revision deadline has passed) am still open to running any clarifying / additional analyses that can be managed in that time, if reviewers consider them pertinent or relevant to their scores.

Much obliged to everyone for their patience and participation!

---

### Meta-Review · Area_Chair_qz8P · 2024-12-16

**Metareview:**

This paper studies the alignment between visual cortex activity and vision/language models, and investigate the idea that high-level visual representations are language-aligned. After rebuttal, it received scores of 5666. On the one hand, reviewers agree that the authors did a good job on experiments and results analysis, and the topic itself is of concrete scientific interest rather than just an engineering paper. On the other hand, several concerns still remain, including (1) the NSD dataset itself mainly used for experiments is limited due to its small dataset size and the use of COCO-like short image captions, (2) potential overclaiming even after paper revision, and (3) its limited machine learning contributions. Overall, the AC would like to recommend rejection by the end.

**Additional Comments On Reviewer Discussion:**

During the rebuttal, one reviewer has increased the score to 5, and the rest kept the scores of 6. Overall, several concerns still remain, and may not be easy to address.

1. More comprehensive experiments can be conducted, such as testing more LLMs. Also, it seems the NSD dataset itself is quite limited, it is suggested to test on more diverse datasets apart from NSD if possible.

2. Even after paper revision, one reviewer mentioned that there is quite a bit of overclaiming in the paper. Three reviewers asked the author to discuss the paper by Yuksekgonul (When and why vision-language models behave like bags-of-words, and what to do about it?), and the authors were not aware of this work and the large volume of related work regarding CLIP text encoders acting like bag-of-word models in their initial revision.

3. The machine learning contributions of this paper are limited.

---

### Decision · Program_Chairs · 2025-01-22

Reject